# Redirecting the Immune Microenvironment in Acute Myeloid Leukemia

**DOI:** 10.3390/cancers13061423

**Published:** 2021-03-20

**Authors:** Stephanie Sendker, Dirk Reinhardt, Naghmeh Niktoreh

**Affiliations:** Department of Pediatric Hematology and Oncology, Clinic of Pediatrics III, University Hospital Essen, 45147 Essen, Germany; stephanie.sendker@uk-essen.de (S.S.); naghmeh.niktoreh@uk-essen.de (N.N.)

**Keywords:** immunotherapy, acute myeloid leukemia, immune-surveillance, microenvironment

## Abstract

**Simple Summary:**

Despite remarkable progress in the outcome of childhood acute myeloid leukemia (AML), risk of relapse and refractory diseases remains high. Treatment of the chemo-refractory disease is restricted by dose-limiting therapy-related toxicities which necessitate alternative tolerable efficient therapeutic modalities. By disrupting its immune environment, leukemic blasts are known to gain the ability to evade immune surveillance and promote disease progression; therefore, many efforts have been made to redirect the immune system against malignant blasts. Deeper knowledge about immunologic alterations has paved the way to the discovery and development of novel targeted therapeutic concepts, which specifically override the immune evasion mechanisms to eradicate leukemic blasts. Herein, we review innovative immunotherapeutic strategies and their mechanisms of action in pediatric AML.

**Abstract:**

Acute myeloid leukemia is a life-threatening malignant disorder arising in a complex and dysregulated microenvironment that, in part, promotes the leukemogenesis. Treatment of relapsed and refractory AML, despite the current overall success rates in management of pediatric AML, remains a challenge with limited options considering the heavy but unsuccessful pretreatments in these patients. For relapsed/refractory (R/R) patients, hematopoietic stem cell transplantation (HSCT) following ablative chemotherapy presents the only opportunity to cure AML. Even though in some cases immune-mediated graft-versus-leukemia (GvL) effect has been proven to efficiently eradicate leukemic blasts, the immune- and chemotherapy-related toxicities and adverse effects considerably restrict the feasibility and therapeutic power. Thus, immunotherapy presents a potent tool against acute leukemia but needs to be engineered to function more specifically and with decreased toxicity. To identify innovative immunotherapeutic approaches, sound knowledge concerning immune-evasive strategies of AML blasts and the clinical impact of an immune-privileged microenvironment is indispensable. Based on our knowledge to date, several promising immunotherapies are under clinical evaluation and further innovative approaches are on their way. In this review, we first focus on immunological dysregulations contributing to leukemogenesis and progression in AML. Second, we highlight the most promising therapeutic targets for redirecting the leukemic immunosuppressive microenvironment into a highly immunogenic environment again capable of anti-leukemic immune surveillance.

## 1. Introduction

Acute myeloid leukemia (AML) is a heterogeneous hematologic malignancy that originates from transformed myeloid precursor cells arising from a hijacked bone marrow microenvironment (BMM). Leukemogenesis is characterized by uncontrolled clonal proliferation of malignant leukemic cells (blasts) that have lost the ability of proper differentiation at various stages of maturation. Our knowledge today suggests that leukemic blasts transduce the surrounding BMM into a leukemia-supportive niche and vice-versa, pointing at a bidirectional crosstalk between leukemic blasts and BMM reciprocally supporting further disease progression [1]. In adults, AML represents the most common form of acute leukemia whilst in pediatrics, it accounts for 20% of all childhood leukemias with an overall survival of about 70% that ranges from 60% to 90% depending on the risk profile [2,3]. However, the prognosis is still poor in cases of refractory disease and relapse, which occur in about 30% of the patients [4,5].

Considering its heterogeneous characteristics, treatment of pediatric AML is adapted to different risk groups, stratified based on different genetic, cytogenetic, and clinical properties. Primarily though, treatment in all groups consists of intensive chemotherapeutic regimens with severe systemic side effects, emphasizing the urgent need for more tolerable, less toxic, and highly efficient treatments. Stepping towards this goal, numerous research works have uncovered substantial mechanisms underlying leukemogenesis and provided pivotal knowledge regarding the biology of AML, paving the way for identification of promising novel therapeutic approaches [6]. However, some of the targeted therapeutic attempts failed to approve desired efficiency and safety in early phase trials and only a few have entered the clinic (examples regarding antibody-based immunotherapies [7,8,9,10] and regarding immune-checkpoint-inhibitor therapies [11,12,13]). Encouraged by the graft-versus-leukemia (GvL) effect following allogenic hematopoietic stem cell transplantation (HSCT) in liquid cancers and the reported success of immunotherapy in solid tumors, immunological treatment opportunities have gradually gained attention [14,15]. Allo-HSCT is one of the oldest and best-known immunotherapies for AML. It has been proven capable of eradicating the residual disease and preventing relapse after the failure of first-line treatment in high-risk patients. The efficacy of HSCT is, however, limited by the severe chemotherapy-related toxicities during conditioning, in acute or chronic graft-versus-host disease (GvHD), or in the event of relapse. Although AML is historically known as an immuno-responsive disease, leukemic blasts reside in a highly supportive, immunosuppressive environment where they adapt various strategies to evade immune surveillance. To date, major efforts have been made to develop new ways to uncover hidden leukemic blasts and to restore intrinsic anti-leukemic immuno-surveillance. Concerning the impact of BMM, which has been shown to be immunosuppressive in AML, a successful immunotherapy should target both the immunologically dysregulated microenvironment and the malignant blasts that can escape the immune surveillance.

In this review, we outline immunosuppressive strategies and the pathophysiological background of disrupted leukemic blasts and microenvironment. In a second step, existing and promising potential immuno-therapeutic approaches are highlighted.

## 2. Acute Myeloid Leukemia Harnesses the Immunological Microenvironment

In addition to oncogenic alterations in hematopoietic cells and BMM, immunological dysregulations contributes to leukemogenesis as well. During the leukemic transition, leukemic stem cells undergo immunoediting, a process that comprises the acquisition of multiple strategies to successfully evade immune surveillance. Consequently, the selected leukemic population is characterized by different immune-evasive mechanisms (Figure 1).

### 2.1. Human Leukocyte Antigens

One known mechanism of immune tolerance in AML was initially observed in patients with relapsed AML following mismatched HSCT, i.e., partially human-leukocyte-antigen (HLA)-incompatible HSCT, which contributes to downregulation of the mismatched HLA class I and II in AML blasts. Upon pressure from the transplanted immune system, this strategy confers ‘survival advantage’ to outgrowing immune-resistant mutant AML clones, characterized by genomic loss of the mismatched histocompatibility determinants. Leukemic blasts evade alloreactive donor T-cell-recognition and killing through this genomic loss of mismatched HLA haplotype which hampers the GvL effect following allogenic HSCT; thus paving the way for leukemia relapse [16,17,18,19,20].

The non-classical HLA class I molecule HLA-G physiologically suppresses the immune system by direct inhibition of dendritic cells (via the inhibitory receptors immunoglobulin-like transcript (ILT)-2 and ILT-4), T-cells (via ILT-2), natural killer (NK)-cells (via ILT-2 and the killer-immunoglobulin-like-receptor (KIR)-2DL4), and monocytes (via ILT-2) [21]. HLA-G has been explored in multiple cancers and its biological and clinical impact as a possible checkpoint inhibitor has been carefully studied [22]. It has been shown that the soluble isoforms of HLA-G are increased in distinct AML samples, especially in monocytic lineages, following interferon (IFN)-gamma and Granulocyte-Macrophage Colony Stimulating Factor (GM-CSF) stimulation [23,24]. In addition, different isoforms of HLA-G have been reported to be secreted or expressed by leukemic cells associated with a higher blast percentage in bone marrow, decreased T cell number, and relapses, which may indicate HLA-G as an additional strategy for AML blasts to evade immune surveillance [25,26]. However, the clinical impact of HLA-G in leukemogenesis appears to be controversial as opposing results do not confirm the clinical implications of HLA-G [27]. In particular, one report did not detect HLA-G in leukemic samples [28].

### 2.2. Checkpoint Molecules

To evade immune surveillance, leukemic blasts (relapse/refractory rather than newly diagnosed patients) express programmed-cell-death ligand-1 (PD-L1), a checkpoint marker that compromises cytotoxic T cells expressing PD-1 [29]. Blockade of PD-L1/PD-1 restores functionality to the exhausted cytotoxic T-cells while synergistically debilitating Treg-mediated immunosuppression, which leads to decreased tumor-burden in an adoptively transferred AML-murine model [30]. Additionally, inhibition of the co-expressed checkpoint-marker T-cells immunoglobulin-mucin 3 (Tim-3) on leukemic blasts generated an even stronger anti-leukemic effect in an AML-murine model [31]. Binding between Tim-3 on leukemic cells and its ligand galectin-9, which is highly expressed in AML blasts, promotes self-renewal via stimulatory β-catenin and NFkB-signaling, and reduces the release of pro-inflammatory cytokines resulting in NK- and T-cell dysfunction [32,33]. In addition, the inhibitory checkpoint-marker C-type lectin-like inhibitory-receptor (CTLA)-4, that competes with CD28, binding CD80/CD86 on leukemic blasts and lymphocyte-activating gene (LAG)-3 has been detected upregulated in primary AML samples. These markers contribute to poor outcome, especially when concurrently expressed in patterns with PD-L1 and/or PD-L2 on leukemic cells [31,34].

### 2.3. T-cellular Immune Dysregulation

In the leukemic microenvironment, T-cells are proposed to reveal altered functional and phenotypic profiles, thereby contributing to immune-suppressive surroundings [35,36,37]. Consistently, the proportion of inhibitory CD4+CD25+ T-regulatory (Treg) cells has been found markedly increased in AML-patient-derived peripheral blood samples and these Treg cells were shown to reduce T-cell proliferation and cytokine production, i.e., IFN-gamma and interleukin (IL)-2 [30,38,39]. In the context of T-cell immunity, the cytokine IL-2 has been proposed to exert two-sided regulative effects. On the one hand, IL-2 supports leukemia-directed lymphocytes, on the other, IL-2 simultaneously favors Treg cells driving the immune-suppressive leukemic microenvironment [40,41]. It has been concluded that besides cell-to-cell contacts, secretion of the immunoinhibitory factors IL-10 and transforming growth factor-beta (TGF-β) contribute to T-reg-mediated suppression of T-cell proliferation [39]. Notably, leukemic cells expressing high indoleamine 2,3-dioxygenase (IDO) were associated with elevated Treg cells [42], probably because of the finding that IDOs induce T-cell conversion into suppressive Treg cells [43]. In addition, AML cells overexpressing the inducible T-cell-co-stimulator ligand (ICOSL) also evoke T-cell conversion, driving the expansion of Treg cells that secrete increased levels of IL-10 and, thereby, favor proliferation and stemness in AML blasts [44]. Hence, AML cells themselves apparently engender T-cell tolerance; thus promoting leukemia progression. In addition, the immunomodulating mediator TGF-β also converts T-cells into Treg cells [45].

Apart from Treg-cell-mediated T-cell suppression, leukemic cells have also been proposed to orchestrate arginase and STAT-3 pathways to reduce T-cell proliferation, which was restored after selective inhibition, confirming the immune-suppressive impact of these mediators [35,38]. Inhibition of cytokine production and cell cycle entry of T-cells exposed to leukemic cell-derived supernatant has been explained by secreted proteins affecting cellular pathways [46]. Intriguingly, previous reports analyzing the effect of AML-cell supernatant on stimulated lymphocytes revealed that cytotoxicity was not affected even though proliferative capacity of exposed T-cells was inhibited [47]. Schnorfeil et al. observed neither inhibition of T-cell proliferation nor activity in AML samples at different stages of disease and, consequently, reasoned that T-cell based immunotherapies may have favorable prospects in AML [48]. More specifically, leukemia antigen-directed T-cell responses have been suggested to be increased when the leukemia burden is minimal due to the investigation that immunocompetent mice transplanted with MLL/AF9-leukemia showed spontaneous antigen-specific T-cell response when a minimum number of leukemia initiating cells were injected, whereas T-cell immunity was exhausted in mice with advanced leukemia [49].

In conclusion, as suggested by numerous research works [35,36,37,46,47,50,51,52], T-cell immunity appears to be compromised in the leukemic microenvironment and an increased proportion of inhibitory Treg cells importantly contribute to this. Nevertheless, data are still limited and exact functional status as well as underlying mechanisms have largely remained obscure.

### 2.4. NK Cell-Related Strategies of Immune Evasion

Multiple studies provide evidence of deregulated anti-leukemic NK-mediated cytotoxicity in AML [53,54,55,56,57,58,59]. Leukemic blasts have been revealed to reduce NK activity by different mechanisms. Sera samples taken from AML patients were enriched in microvesicles containing increased levels of TGF-β, which led to reduced anti-leukemic NK-cytotoxicity in vitro [53]. Conversely, in an NK cell—AML-blasts co-culture system, Stringaris et al. detected an increased level of IL-10 but not TGF-β, suggesting that inhibition of NK cells by AML blasts depends on IL-10 instead [58]. Myeloid-derived suppressor cells (MDSCs) are a distinct immature cell population that facilitates immune-evasive strategies [60]. Recently, MDSCs were shown to suppress the anti-leukemic activity of NK cells, mediated by IDO and prostaglandin-E2 (PGE2) and exosomes [61]. AML blasts are supposed to constitutively express IDO, nonetheless, the exact contribution to AML progression and immune tolerance remains to be elucidated [62]. Leukemic blasts induce MDSC proliferation and differentiation into tumor-associated macrophages, which further inhibit immunogenicity favoring their survival; thus driving leukemia progression resulting in deteriorated outcomes [60,63,64]. Deficient NK cell function can be traced back to weak expression of different activating natural cytotoxic receptors (NCRs) on AML-derived NK cells [54,55,56]. Since a small number of AML-derived NK cell samples showed diminished anti-leukemic activity regardless of expressing high NCR-levels, it is suggested that in some cases, AML cells express a low level of NCR ligands to escape NK-mediated cytotoxicity [55,57]. NK- and T-cells also recognize cancer cells through the activating receptor NK-activating surface marker (NKG2D)-binding to its ligands MHC class I-related chain (MIC) -A and -B and UL16-binding protein (ULBP)-1 and -2 [65]. In acute leukemia, expression of NKG2D is downregulated or not present at all [57,66]. Leukemic cells contribute to this immune-evading strategy since they release increased levels of TGF-β [53,67]. Moreover, secretion of soluble NKG2D-ligands by leukemic blasts inhibits NGK2D expression on NK cells, resulting in decreased anti-leukemia activity [68]. Of note, stem-cell-like subsets of AML blasts that lack surface-bound NKG2D-ligands (attributed to elevated poly-ADP-ribose-polymerase-1 (PARP1)) efficiently evade NK-driven immune control, conferring a selective advantage in the absence of NKG2D ligands [69]. Impaired binding of perforin is another resistance mechanism through which distinct AML blasts can elude perforin-mediated NK cell-lysis [70]. Mesenchymal stem cells decrease NK-cytotoxicity and IL-2-induced NK cell expansion by secretion of IDO and the cyclooxygenase (COX)-2 product, prostaglandin E2 (PGE2) [71]. In addition, PGE2 has been revealed to suppress IL-15-stimulated NK cell reactivity, in terms of cytotoxicity and IFN-gamma egress [72]. Furthermore, proliferation and differentiation of CD8+ T-cells are reported to be inhibited by diminished tryptophan levels, owing to enzymatic digestion by IDO [73] and its catabolite was shown to decrease NK-activating surface marker (NKG2D) [74]. Also, the enzymatic activity of IDO, highly expressed in BM stromal cells causes the conversion into Treg cells [43,75] and boosts their immune-suppressive function [76,77]. In addition to the before-mentioned Treg-cell-mediated T-cell suppression, Treg cells have been explored to actively reduce NK cell cytotoxicity and NKG2D receptor expression through membrane-bound TGF-β [78]. These results may imply the high value of TGF-β, PGE2, and IDO disturbing the immunogenic anticancer response. However, more research is required to explore the decisive impact of immunological dysregulating factors in the BMM, especially in AML.

## 3. Therapeutic Approaches Redirecting Immuno-Suppressive Microenvironment

As the immunological microenvironment is hijacked by leukemic blasts leading to these malignant cells evading the immune surveillance, multiple novel therapeutic approaches target immune-evasive strategies to restore anti-leukemic immune activity (Figure 2).

### 3.1. Checkpoint Inhibitors

In AML, expression of checkpoint molecules, such as PD-1 and CTLA-4 by immune cells or PD-L1 and Tim-3 by leukemic blasts has been reported as a sufficient strategy to escape immune surveillance. Clinical trials evaluating immune checkpoint-inhibitor (CPI) targeting PD-1 (Nivolumab, Pembrolizumab) and CTLA-4 (Ipilimumab) achieved encouraging success in solid tumors [79]. Feasibility and safety of these CPIs are currently being explored in different ongoing clinical phase I/II trials (reviewed in [80]), (Appendix A).

Treatment of recurrent hematologic malignancies after allogenic HSCT with the monoclonal CTLA-4 inhibitor Ipilimumab achieved objective clinical response results (i.e., Graft versus Malignancy, GvM) in less than 10% of cases; however, Ipilimumab treatment did not lead to GvHD whilst tolerable immune-mediated adverse events occurred in 4 of 29 patients (14%) with a positive treatment response [12]. In a comparable setting, evaluating the response to Ipilimumab after HSCT, relevant immune-related adverse events were seen in 10 out of 28 cases. Among others, one patient died, and 4 patients developed dose-limiting toxicity due to GvHD. Complete remission occurred in 4 out of 12 cases [13,81]. A retrospective analysis across different trials of AML and high-risk MDS patients treated with PD-1 and PD-L1 inhibitors observed comparable dose-limiting immune-associated adverse events [82]. Incidence of adverse events and objective response to CTLA-4 blockade appears to be dose-dependent and response to treatment has been linked to histological subtypes of malignancies as a durable remission was noted in the enrolled extramedullary AML cases including three with leukemia cutis who also developed a mild GVHD [12,13]. The Anti-PD-1 antibody, CT-011, when administered as a single-agent, demonstrated tolerable safety but limited anti-leukemic efficiency in patients with advanced AML [83]. The previously mentioned immune-related adverse events (i.e., GVHD) as well as potential damage due to treatment-related toxicity should be carefully weight up with the effectiveness of CPI. In addition, it has been proposed that limited clinical success of CPI in AML could be due to heterogenic levels and distinct patterns of checkpoint-molecule expression in AML [34]. Investigations in a murine model observed increasing co-expression of checkpoint molecules during AML-progression, resulting in an exhausted anti-leukemic effector cells response [31].

To overcome this hurdle, which could be a major cause for modest response rates of CPI monotherapy, combinatory treatment approaches are currently under investigation, striving for better patient outcomes. Studies combining CPIs with chemotherapy in AML are currently under investigation (NCT03417154, NCT02768792, NCT02464657). Preliminary results from a phase II trial assessing additional administration of Nivolumab three weeks after standard chemotherapy (anthracycline and cytarabine) in adult patients, including 42 newly diagnosed AML and 2 high-risk MDS demonstrate efficacy and feasibility, of note 6 out of 44 patients had grade 3/4 immune-related toxicities [84]. Moreover, Nivolumab combined with the epigenetic regulator Azacytidine has been determined as an efficient and well-tolerated therapy-option in relapsed, high-risk AML patients in a phase I/II study [85].

Effective CPI therapy requires proper effector lymphocyte reactivity, which is reduced in the immunosuppressive leukemic BMM, thus presumably hampering therapeutic effectiveness. Recognition by T-cells could be further reduced due to a relatively lower tumor mutation burden in AML, which has been determined as predictive response marker for CPI therapy [86]. Considering the highly immunosuppressive surrounding as well as the stated low immune-checkpoint-expression baseline in AML [86,87], which presumably hamper efficaciousness of CPI administration of hypomethylating agents (HMA), which promote anti-leukemic immune-response by upregulation of cellular reactivity and checkpoint-molecule expression [88,89] could be a valuable approach to ameliorate therapeutic response to CPI in AML Combined checkpoint blockades inhibiting PD-(L)1 and CTLA-4 or Tim-3 administered together with HMA are currently under investigation (NCT02530463, NCT03066648). Among these checkpoint molecules, targeting Tim-3/Gal-9 could be a promising therapeutic approach as this has been proposed to specifically promote LSC self-renewal [32,90] and to be upregulated in therapy failure in AML [91].

Since different studies revealed distinct co-expression patterns of immune checkpoints with a prognostic impact [31,34,92], dual or combined checkpoint inhibition might be an attractive future approach to stop immune evasion. In a pediatric refractory AML patient, the concurrent blockade of CTLA-4, PD-L1, and HMA initially improved symptom control without adverse events, but ultimately was not able to stop the lethal progression. To assess clinical efficacy of this combinatorial approach in pediatric AML further investigations are warranted [93].

### 3.2. Antibody-Based Therapy

Despite the paucity of truly tumor-specific antigens, leukemia-associated surface molecules preferentially expressed by AML-blasts were developed as potential therapeutic targets.

CD33 and CD123 are partly expressed on healthy HSCs as well as most AML blasts, including LSCs [94,95,96,97]. Targeting these and other promising leukemic antigens using un-modified monoclonal antibodies failed to achieve hoped-for success [8,9] and reviewed in [10]. Recent studies focused on engineered antibodies equipped with a toxic payload, referred to as antibody–drug conjugates (ADC), or with an Fc-receptor engineered to increased CD16 affinity to potentiate antibody-dependent cell cytotoxicity (ADCC).

Gemtuzumab ozogamicin (GO), a CD33-directed ADC loaded with cytotoxic calicheamicin, showed feasibility and efficacy in early clinical studies conducted on a compassionate-use basis in pediatric patients with relapsed/refractory AML [98,99]. GO is approved by the U.S. Food and Drug Administration (FDA) for adults with CD33-positive AML as well as adults and children, aged 2 years and older with relapsed/refractory (R/R) AML. The European Medicines Agency’s (EMA) approval of GO for newly diagnosed AML patients, older than 15 years was based on superior outcome in the randomized phase 3 study, ALFA-0701 (NCT00927498) [100]. Recently the GO indication has been expanded by the FDA for children, older than 1 month with de-novo CD33-positive AML based on data of the randomized phase III study AAML0531 (NCT00372593) [101]. Recent results of a phase I trial evaluating CD123-directed ADC (coupled to an alkylating agent) in relapsed/refractory AML revealed safety and clinical responses [102], (Appendix A).

Whilst CD33 and CD123 are also expressed on subsets of HSCs [97,103], CLL-1 is widely expressed on AML blasts and LSCs but absent on HSCs; thus presenting a promising anti-leukemic target-sparing hematopoietic regeneration [104,105]. In vitro and in vivo AML models revealed compelling anti-leukemic efficacy for CLL-1-directed ADC and for the T-cell-engaging bispecific antibody αCLL1-αCD3, which was superior to CD33-directed drugs since CLL-1 spares HSCs [106,107].

Preliminary results of a phase I study evaluating a dual affinity re-targeting anti-CD123-CD3 antibody in refractory/relapsed AML and MDS reported T-cell-mediated efficacy and acceptable toxicity with cytokine release (grades 1 and 2) as a frequent but manageable adverse event [108]. Interestingly, in an animal study using an AML-monkey model, cytokine release was reduced when the CD3-low affinity variant of bispecific T-cell engagers was administered compared to the CD3-high affinity variant [109].

The Fc-engineered CD123 antibody with increased CD16-dependent NK cell affinity (CSL362) efficiently achieved NK cell-mediated cytotoxicity of AML blasts and LSCs in preclinical investigations [110,111]. However, in clinical phase I/II trials, CSL362 failed to prove this anti-leukemic potency, which presumably is due to lack of NK cells in heavily pretreated patients with advanced AML [112,113]. CD123- and CD33-directed bi- and tri-specific NK cell engagers (CD33×CD16, CD123×CD33×CD16, and CD123×IL15×CD16) could boost ADCC [114,115,116]. An NK-mediated anti-leukemic activity can be supported by the administration of HMA. This has been shown to partly eliminate immune-evading strategies, i.e., by upregulation of activating NK-receptors and reenforced NKG2D-mediated NK activation resulting in increased NK activity [89,117]. A new concept unites immuno-CPI and T-cell engagers to restore exhausted T- and NK cell activity and, thus reinforce the efficiency of bispecific T-cell engagers (BiTEs) [118]. This raises the notion that immunotherapeutic agents together with NK cell engaging agents could also be a suitable future opportunity, but these hypothetical approaches need to be investigated.

For treating solid cancers, tetra-specific antibodies were developed, consisting of CD16 crosslinked to IL-15 as an NK cell activating moiety fused to single-chain variable fragments (scFvs) binding cancer-associated antigens [119]. Regarding genetic and morphological heterogeneity of AML blasts, this multi-specific approach could be a conceivable alternative antibody-based therapeutic approach in AML to specifically target leukemic blasts based on individual identified surface marker profile.

### 3.3. Cell-Based Therapy

Initially, cellular therapeutic approaches were developed to target leukemia inspired by powerful T-cell-mediated anti-leukemic immunity in the setting of allogenic HSCT [120]. The infusion of haploidentical T-cells in patients with relapsed AML in a single center study for over two decades, demonstrated moderate feasibility which was limited by immune-dependent toxicity based on the severity/risk of GvHD [121].

In a preclinical setting, anti-leukemic activity has been proven for genetically modified chimeric-antigen-receptor (CAR)-expressing T-cells directed against CD123 or CD33 on leukemic cells [122,123]. Transduced anti-CD33-CAR T-cells are currently under investigation in early phase trials in children and adolescents with relapsed/refractory AML (NCT03971799). In a preclinical in vivo trial, bispecific CAR-T-cells simultaneously targeting CD123 and CD33 efficiently eliminated leukemic blasts and LSCs [124].

However, increased T-cell-mediated toxicity, especially regarding elevated cytokine release and GvHD should be considered when using CAR T-cells [125]. Conversely, NK cells show favorable safety (including reduced cytokine release and diminished GvHD risk) with high feasibility as they are not MHC-restricted. Consequently adoptively transferred NK cells are currently discussed as promising future cost-efficient, off-the-shelf drugs [126,127,128].

So far, different NK-based immunotherapeutic strategies have been developed. Cytokine-stimulated and lentiviral-transduced CAR-engineered killer cells targeting CD123 exhibited safety and anti-leukemic cytotoxicity against leukemic blasts in preclinical investigations [129]. Adoptive cytokine-induced allogenic NK cell transfer already demonstrated anti-leukemic efficacy and safety in a pediatric patient with relapsed AML post-HSCT [130], and further phase 1/2 trials in children with relapsed/refractory AML are currently ongoing (NCT01898793 and NCT03068819). Treatment with CD33-directed CAR-NK cells in a clinical phase I trial was safe but failed to achieve significant anti-leukemic efficacy [128], (Appendix A). 

Transfusion of highly purified, allogenic haploidentical KIR-HLA mismatched NK cells following low-dose immunosuppression (with cyclophosphamide and fludarabine) and administration of IL-2 yielded beneficial response rates at low toxicity without GvHD and has been proposed as an innovative alternative consolidation therapy for pediatric and adult AML patients [131,132]. Compared to allogenic haploidentical NK cell infusion in high doses of 29 × 10^6^/kg, transfusion of a lower dose of 12.5 × 10^6^/kg could not achieve beneficial results under otherwise comparable conditions in children with AML in first complete remission [131,133]. Therefore, this raises questions about the effective dosage. Interestingly, in elderly patients with AML, administration of purified allogeneic mismatched NK cells in much lower concentrations, ranging from at least 1 × 10^6^ to 5 × 10^6^ /kg was found to be clinically efficient and feasible [134]. Therefore, it is likely that additional variables influence the response, such as preparation technique due to purification but also in vivo stimulation by immune-activating cytokines [135].

### 3.4. Therapeutic Impact of Cytokines and Immune-Modulating Factors

Since different pro-inflammatory cytokines, such as IL-12, -15, -18, and -21 were shown to reinforce leukemia-directed immunogenicity [136,137], cytokine-based therapies have been tested alone or in combination with other immunogenic anti-leukemic approaches.

To maximize anti-leukemic immunogenicity, ex vivo stimulation using a mix of IL-12, IL-15, and IL-18 provokes differentiation into memory-like NK cells, which exhibit enhanced leukemia-directed reactivity compared to non-activated/native NK cells [135,138]. These cytokines also sensitize NK cell-bound IL-2 receptors, thus, ameliorating NK cell functionality [139]. However, IL-2 stimulates immunosuppressive Treg cells rather than NK cells according to the abundant expression of high-affinity IL-2 receptor alpha-chain (IL-2Rα and CD25), which prevents IL-2 from stimulating NK cells [135]. Therefore, several attempts testing IL-2 as a single-drug failed to achieve anticipated anti-leukemic activity [140,141]. Results of a meta-analysis assessing the outcome of IL-2 as a single agent showed that it was not superior over no intervention regarding overall and disease-free survival [142].

Different strategies have been developed to bypass this obstacle. To promote effectiveness of allogenic NK cell-based therapy, administration of IL-2 diphtheria toxin fusion protein, that depletes Treg cells bearing IL2Rα (CD25), restored anti-leukemic NK cell immunogenicity when administered prior to NK-infusion and IL-2-administration [135]. In phase I/II trials, use of recombinant IL-15 instead of IL-2 fostered NK cell expansion in vivo and yielded beneficial remission rates in patients with advanced AML (NCT01385423 and NCT02395822) [143]. Of note, these two trials showed that the pharmacokinetic clearance, cytokine storm, and neurotoxicity were linked to subcutaneous administration but not intravenous administration [144]. A phase I trial is currently investigating the safety and feasibility of IL-21 ex-vivo-expanded NK cells [145].

The IL-15/IL-15Ra super-agonist (N-803) is supposed to mimic the immunostimulatory effect exerted by antigen-presenting cells that trans-present IL-15 linked to IL-15Ra to T-cytotoxic- and NK cells through a shared IL-2/IL-15-alfa/beta receptor [144]. Administered as single drug in early clinical trials, N-803 was well-tolerated and effective in anti-leukemic disease-control [146]. Recently, further ongoing trials evaluated administration of N-803 in combination with NK cell infusion (NCT01898793, NCT02782546), (Appendix A).

Even though class one interferons are proposed to exercise anti-leukemic and pro-immunogenic functions on malignancies, utility as a potent therapeutic opportunity was modest when investigated as sole treatment [147,148].

Considering the complexity of the leukemic BMM, various further immunosuppressive factors also restrict anti-leukemic immunogenicity. These limitations could be offset using drugs that alleviate leukemia-driven lymphocyte suppression. In this regard, additional suppression of inhibitory mediators TGF-β or IDO 2,3 may present promising approaches to overcoming leukemic restrictions and increasing the success of cell-based therapy in AML.

Inhibition of IDO 2,3 was found to promote cellular T- and NK cell-mediated immunoregulation and to inhibit Treg-cell conversion [62,75]. Antagonizing PGE2 also resulted in improved anti-leukemic reactivity [72]. Therefore, additional targeting of these immunoregulative approaches may increase the efficacy of cell-based therapy.

Since research explored the immuno-suppressive role of TGF-β in different cancers, antagonizing TGF-β reinforces the lymphocyte-mediated immune response in fighting malignant disease, as has been shown for non-hematologic malignancies [149]. In this regard, preclinical experiments describe that the TGF-β inhibition sustains anti-leukemic NK cell reactivity despite exposure to a pathological level of TGF-β [150]. 

Increased levels of cell-bound and soluble HLA-G are overexpressed in distinct AML subtypes and presumably present a highly efficient approach to restoring immune response as HLA-G regulates NK, T-, and dendritic cells. In addition, HLA-E also represses NK cell functionality binding NKG2A, thus presenting another potential target. To date, preclinical and clinical data focusing on the feasibility of stated approaches are limited or even do not exist.

### 3.5. Vaccines

Vaccination has attracted rising attention as a potential therapeutic tool to facilitate anti-leukemic immunological surveillance. To provoke leukemia-directed immune response, two main vaccine-based strategies are focused on (i) leukemia-associated antigens and (ii) antigen-presenting dendritic cells, either stimulated by AML cells, used as fusion-product or genetically modified.

Wilms tumor gene (WT-1) peptide-based vaccines (containing HLA-A*2402-restricted, natural, or modified 9-mer WT1 peptide emulsified with montanide ISA51 adjuvant) were generally well-received and achieved specific cytotoxic T-cell reactivity resulting in reduced leukemic burden with markedly decreased MRD markers in the majority (60%) of treated AML patients after consolidation therapy [151].

Intracutaneous administration of the WT-1 vaccine is suggested as a safe and efficient promising second-line maintenance therapy strengthening GvL effect post HSCT when given at the lowest point of leukemic load [152]. This could be explained by a lymphocyte-depleted microenvironment that may provide optimal conditions for the expansion of specific leukemia-directed lymphocytes [152].

In phase I/II trials (NCT00665002 and NCT01266083) administration of the multivalent WT-1 vaccine, galinpepimut-S, triggered a specific immune response related to improved survival without causing relevant toxicity in treated AML patients after achieving first complete remission [153]. These encouraging results prompted the initiation of a phase III trial examining efficiency and safety in a larger cohort (NCT04229979).

Dendritic cells (DCs) are considered as the most powerful antigen presenting cells and are therefore ideally suited as cellular adjuvants for therapeutic vaccination. Results from a phase II trial (NCT00965224) suggest WT1-mRNA loaded DCs as an effective strategy to induce antigen-specific T-cell response and subsequently prevent or delay relapse after standard chemotherapy [154], (Appendix A). Vaccination with autologous DCs, electroporated with human telomerase reverse transcriptase (hTERT) encoding mRNA was safe and feasible and connected to a prolonged recurrence-free survival of treated AML patients in complete remission [155]. A noteworthy strategy is a hybridoma of patient-derived DC and leukemic cells harnessed as individualized vaccination, which was found to prime T-cell expansion and specific anti-leukemic reactivity in pre-treated AML patients; thus preventing them from disease relapse [156].

An injectable cryogel vaccination which delivers the immunostimulatory CpG-oligodeoxynucleotide and GM-CSF together with leukemia-related antigens induced a sound anti-leukemic immunity including DC activation which prevented treated AML-bearing mice from leukemic engraftment and eradicated established leukemia when administered together with chemotherapy [157]. Interestingly, this cryogel vaccine, together with chemotherapy achieved eradication of established AML even in the absence of a defined antigens, which has been attributed to accumulated, apoptotic AML blasts expressing enough antigens to boost DC attack [157].

Facing limitations in clinical attempts of new immune-therapeutic strategies and of the intricate immunological leukemic microenvironment, a combination of cellular- and non-cellular-based immunotherapeutic strategies may be strong enough to efficiently overcome leukemic immune-evasion.

### 3.6. Oncolytic Viruses

Oncolytic virus (OV) has been determined as another appealing therapeutic approach to directly kill malignant cells and boost anti-tumorigenic immunity in various malignant diseases [158,159].

Due to a natural tumor-specific tropism, OVs selectively contaminate, proliferate in, and destroy tumor cells without harming healthy cells. Simultaneous release of tumor-specific damage-associated molecular patterns (DAMP) including highly immunogenic tumor-derived peptides alert the immune system and cause anti-tumor immunity re-modifying BMM into a highly inflammatory site [160]. Therefore, OVs may act complementary to other immune therapeutics by attracting the acquired immune-system-enhancing anti-tumor immune response [161].

Several in vitro and in vivo studies evaluated the use of (modified) OVs alone or in combination with other drugs in hematologic malignancies including AML. Measles vaccine virus (MeV) activated IFN signaling and directly lysed AML cells expressing the measle entry receptor CD46, a complement regulatory cofactor. Combination of MeV with leukemia site-specific generation of the cytotoxic active drug 5´flour uracil using modified MeV equipped with super cytosine deaminase (MeV-SCD) additionally intensified this anti-leukemic effect [162].

The engineered adenoviral vector, zA4, coated by TNF-apoptosis-inducing related-ligand (TRAIL) efficiently evokes cytotoxicity and significantly inhibits leukemia-cell proliferation, and additive ginsenoside (Rh2) enforces anti-tumorgenicity by inducing the expression of TRAIL-related receptors on leukemic blasts [163].

UV-light-inactivated herpes simplex virus-1 (UV-HSV-1) exerts cytolysis of AML cells via activation of leukemia-directed immune response and, synergistically with IL-15 and IL-2, induces cytolytic activity of stimulated NK cells, suggesting UV-HSV-1 as a therapeutic supplement to reinforce adoptive cell therapy [164].

Despite the fact that AML cells are resistant to the direct lytic effect of the coxsackievirus strain CVA21, this OV exerts a powerful innate and adaptive immune response, killing leukemic blasts [165]. This could be a rationale for combinatorial treatment of CVA21 with other immunotherapeutic agents to further boost anti-leukemic immunogenicity. Other preclinical data proved direct and indirect cytotoxic efficacy of reovirus as an immunotherapeutic agent that stimulates immune response with enhanced oncolytic NK activity [166].

To further augment treatment efficacy, genetically modified OVs are in development. Transduced OVs, expressing distinct immunogenic TAA, have been assessed to strengthen desirable anti-tumoral immunity [167,168]. Other OVs, that were engineered to synthesize proinflammatory cytokines causing stimulation of the immune system, have already been evaluated in clinical trials [169,170]. One example is the recombinant vesicular stomatitis virus (VSV), encoding IFN-beta and the sodium-iodide symporter (NIS) (VSV-mIFNβ-NIS), which is currently being tested in a Phase I study in patients with recurrent AML (NCT03017820), (Appendix A). Combined with PD-L1 blockade, VSV-mIFNβ-NIS further augments antitumor activity with enhanced infiltrating T-cells reducing leukemia burden in an AML-murine model [169]. OVs armed with bi- and tri-specific T-cell engagers to locally boost T-cell-mediated tumorgenicity are currently under investigation [171].

The variety of modified OVs leads to the hypothetical concept to utilize OV as a supplier of leukemia-specific antigens, potentially addressing the challenging lack of unique targetable antigens in AML. For solid tumors, the combination of CD19 CAR-T cells with engineered OVs delivering CD19 as an amenable target to tumor cells turned out to be safe and effective in preclinical proof of concept trial [172].

Together, OVs appear as an exciting novel immunotherapeutic strategy, that needs to be followed-up in further investigations. Clinical trials in AML are pending to determine the most potent viral strain and combinatorial immunotherapeutic setting for the efficient killing of AML blasts in pre-treated patients.

## 4. Conclusions

In AML, immunotherapeutic approaches have recently emerged as a promising step towards the long-term goal of an improved, more specific anti-leukemic treatment. Findings of disrupted immunoregulation in AML, that contributes to leukemogenesis and promotes disease progression led to innovative therapeutic attempts to reconstitute anti-leukemic immunosurveillance. Promising results of preclinical and clinical trials point at significant therapeutic values of oncolytic viruses, vaccines, and cytokines as well as cellular- and antibody-based immunological approaches. Combining these different immunotherapies could increase treatment efficacy and mark future directions towards a new generation of successful, targeted therapy in AML. However, clinical success may be limited by possible adverse toxic side-effects. To enhance safety and therapeutic effectiveness, a deeper understanding of immune dysregulation in AML is urgently needed. This may further enrich our repertoire of potent immunological drugs to eliminate leukemic blasts in AML.

## Figures and Tables

**Figure 1 cancers-13-01423-f001:**
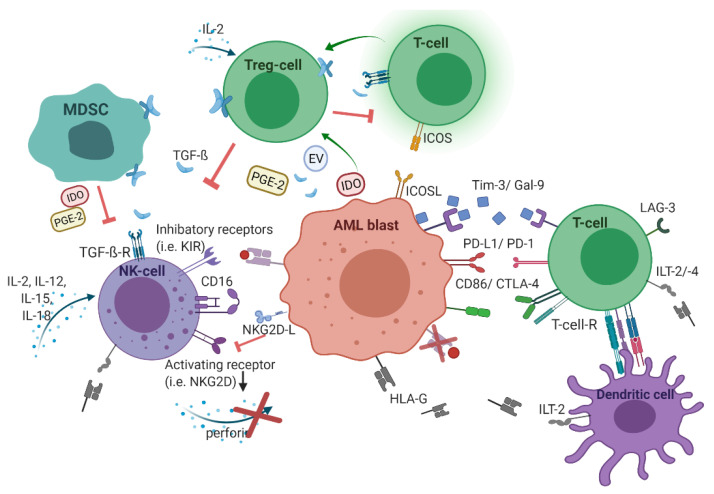
The immunological microenvironment in acute myeloid leukemia (AML). AML blasts reduce antigen-presentation through downregulation of classical human-leukocyte-antigen (HLA)-presentation. Non-classical HLA-G is supposed to suppress immunogenicity. Checkpoint molecules promote immune evasion (Gal-9/Tim-3, PD-L1/PD-1, CD86/CTLA-4, and LAG-3). Secretion of TGF-β and indoleamine 2,3-dioxygenase (IDO), as well as inducible T-cell-co-stimulator ligand (ICOS)/ICOS-ligand interplay induces T-cell conversion into immunosuppressive T-regulatory cells (Treg) cells. Myeloid-derived suppressor cells (MDSC) suppress natural killer (NK)-cell-mediated cytotoxicity, i.e., via IDO, prostaglandin-E2, and TGF-β. Figure 1 was created with biorender.com (accessed on 20 February 2021).

**Figure 2 cancers-13-01423-f002:**
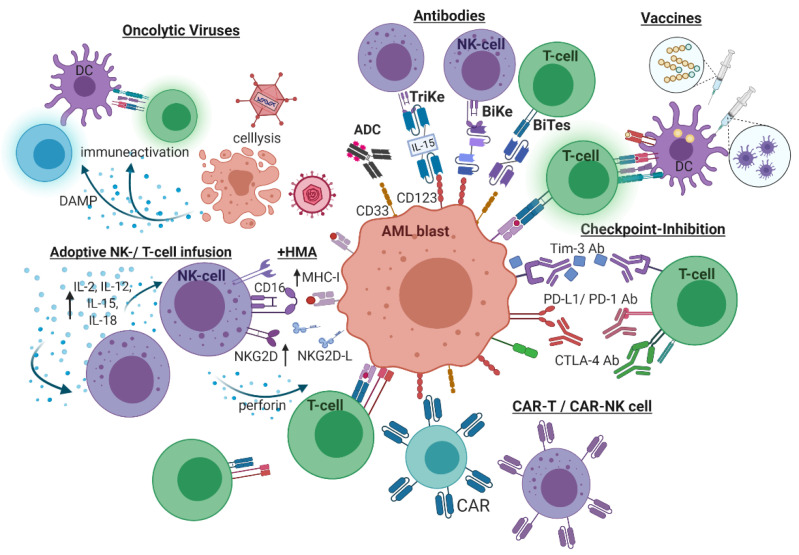
Immunotherapeutic approaches in AML. Checkpoint blockade prevents immune-suppressive signaling through programmed cell-death protein-1/programmed cell-death ligand (PD-1/PD-L1), cytotoxic T-lymphocyte associated protein 4 (CTLA-4), T-cells immunoglobulin-mucin 3 (Tim-3)/galactin-9 (Gal-9), as prominent examples. Cellular-based immunotherapeutic approaches comprise adoptive T- and NK cell infusion. Exposure to activating cytokines (IL-2, -12, -15, and -18) augments NK-mediated cytotoxicity. Additional administration of hypomethylating agents (HMA) potentiate cellular immune response. Chimeric antigen receptor (CAR) is another strategy to improve leukemia directed T- and NK cell reactivity. Leukemia-associated antigen-directed antibodies stimulate antibody-dependent cell-mediated cytotoxicity (ADCC). Antibody–drug conjugates (ADC) are linked to cytotoxic agents to directly lyse targeted leukemic blasts. Bispecific T-cell engager (BiTE), bi- and tri-specific NK cell engager (BiKE, TriKE) bind and crosslink leukemic antigens to T- and NK cells facilitating anti-leukemic reactivity. Vaccine therapy can be either based on antigens or dendritic cells, presenting neoantigens to T-cells resulting in leukemia-directed cytotoxicity. Oncolytic viruses directly lyse infected AML cells and specifically reinforce anti-leukemic immunogenicity, i.e., through released specific damage-associated molecular patterns (DAMP). Figure 2 was created with with biorender.com (accessed on 20 February 2021).

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
