# Peer review of "Redirecting the Immune Microenvironment in Acute Myeloid Leukemia"

_cancers, 2021, doi:10.3390/cancers13061423_

Round 1

Reviewer 1 Report

A comprehensive review of the present knowledge about immunological mechanisms in the pathogenesis of AML and possible new treatment modalities. Very well written and thouroughly done work, nothing to add or change. It was a great pleasure to read this.

Author Response

Reviewer 1

Review Report (Round 1)

A comprehensive review of the present knowledge about immunological mechanisms in the pathogenesis of AML and possible new treatment modalities. Very well written and thoroughly done work, nothing to add or change. It was a great pleasure to read this.

Response: Thank you!

Reviewer 2 Report

In this manuscript, Sendker and colleagues review current knowledge about the immune microenvironment in AML as well as immunotherapeutic strategies investigated in the setting of AML, spanning both pediatric and adult AML as well as both non-transplanted and post-allogeneic therapeutic settings. Due to this very broad approach, the manuscript necessarily lacks depth in many parts, but provides a good general overview.

My main concern is that the very brief summaries of published results or ongoing clinical trials often leave the reader unclear about the setting, specifically pediatric vs. adult and non-transplanted vs. post-allo. This needs to be improved throughout the article.

Besides, I have the following specific comments:

  • Line 10: Should this be “restricted by dose-limiting therapy-related toxicities”?
  • Line 86: The actual “known mechanism” should be named in the title or more in the beginning of the paragraph, otherwise it is unclear to the reader what the authors are referring to.
  • Line 225: What cells is this sentence referring to with respect to checkpoint molecule expression? Leukemic cells or immune cells?
  • Line 227: Pembrolizumab is not directed against PD-L1, but PD-1.
  • Line 228: Stating that checkpoint inhibitors have achieved “encouraging initial results in solid tumor” seems to be a really massive understatement.
  • Line 239: A retrospective analysis of patients across different trials should not be called a “trial conducted in AML and high-risk MDS patients”.
  • Line 253: This was a phase II trial, not phase I.
  • Line 254: Nivolumab was not administered simultaneously, but 3 weeks after standard chemotherapy.
  • Line 255: Several patients were reported with G3/4 adverse events. This is not reflected in the statement “without severe adverse events”.
  • Line 259ff: Referencing only a case report seems to be insufficient for the statements in this paragraph.
  • Line 270ff: The context of this short stand-alone paragraph seems unclear.
  • Line 288ff: Gemtuzumab Ozogamicin (please correct the typo!) is an approved drug for AML. This information should be added to this paragraph that only refers to compassionate use.
  • Line 297ff: There seems to be a slight misunderstanding what BiTEs (the name belongs to the proprietary Amgen platform) are, as the authors are repeatedly referring to a BiTE when other bispecific antibody constructs are described.
  • Line 321ff: It is unclear to me what the sentence in its current form is supposed to say.
  • Line 426: It is not clear what kind of WT-1 vaccines the authors are referring to here.
  • Line 440ff: Compared to other sections, DC-based vaccines are very insufficiently reviewed here. From this paragraph, it appears that the hybridoma strategy with a single clinical trial seems to be the only approach. Please consider also to mention strategies based on monocyte-derived DCs, e.g. Khoury, Cancer 2017, Anguille Blood 2017, Lichtenegger, Clin Transl Immunology 2020.
  • References: It would be helpful for the reader if conference abstracts would be marked as such compared to full publications.

Author Response

Reviewer 2

Review Report (Round 1)

In this manuscript, Sendker and colleagues review current knowledge about the immune microenvironment in AML as well as immunotherapeutic strategies investigated in the setting of AML, spanning both pediatric and adult AML as well as both non-transplanted and post-allogeneic therapeutic settings. Due to this very broad approach, the manuscript necessarily lacks depth in many parts, but provides a good general overview.

My main concern is that the very brief summaries of published results or ongoing clinical trials often leave the reader unclear about the setting, specifically pediatric vs. adult and non-transplanted vs. post-allo. This needs to be improved throughout the article.

Response:

Thank you for your thoughtful suggestions. We have created a tabular overview, which summarizes the main important data including the setting of selected trials.

Besides, I have the following specific comments:

  • Line 10: Should this be “restricted by dose-limiting therapy-related toxicities”?

Response: We have edited as suggested.

  • Line 86: The actual “known mechanism” should be named in the title or more in the beginning of the paragraph, otherwise it is unclear to the reader what the authors are referring to.

Response: We have added the “known mechanism of immune tolerance in AML”.

  • Line 225: What cells is this sentence referring to with respect to checkpoint molecule expression? Leukemic cells or immune cells?

Response: We have specified as requested.

  • Line 227: Pembrolizumab is not directed against PD-L1, but PD-1.

Response: Important advice. We have corrected the typo.

  • Line 228: Stating that checkpoint inhibitors have achieved “encouraging initial results in solid tumor” seems to be a really massive understatement.

Response: We have modified as requested.

  • Line 239: A retrospective analysis of patients across different trials should not be called a “trial conducted in AML and high-risk MDS patients”.

Response: We have modified as requested.

  • Line 253: This was a phase II trial, not phase I.

Response: We have corrected as requested.

  • Line 254: Nivolumab was not administered simultaneously, but 3 weeks after standard chemotherapy.

Response: We have modified as requested.

  • Line 255: Several patients were reported with G3/4 adverse events. This is not reflected in the statement “without severe adverse events”.

Response: We have modified the statement.

  • Line 259ff: Referencing only a case report seems to be insufficient for the statements in this paragraph.

Response: Important point. We agree with you and assume that further investigations are warranted to assess clinical efficacy of this combinatorial approach in pediatric AML. We have modified the corresponding statement.

  • Line 270ff: The context of this short stand-alone paragraph seems unclear.

Response: We have modified the first part of this stand-alone paragraph and incorporated it in a previous sentence regarding limitation of CPI in AML and how to overcome these using HMA in combinatorial approaches. The second part was omitted, as there was no appropriate context.  

  • Line 288ff: Gemtuzumab Ozogamicin (please correct the typo!) is an approved drug for AML. This information should be added to this paragraph that only refers to compassionate use.

Response: Thank you for the advice, we have corrected the typo and added a short statement on recent updates of the approval of Gemtuzumab Ozogamicin by the FDA and EMA.

  • Line 297ff: There seems to be a slight misunderstanding what BiTEs (the name belongs to the proprietary Amgen platform) are, as the authors are repeatedly referring to a BiTE when other bispecific antibody constructs are described.

Response: We agree and corrected as requested.

  • Line 321ff: It is unclear to me what the sentence in its current form is supposed to say.

Response: We have rewritten the sentence to clarify the meaning.

  • Line 426: It is not clear what kind of WT-1 vaccines the authors are referring to here.

Response: We have specified the kind of WT-1 vaccine.

  • Line 440ff: Compared to other sections, DC-based vaccines are very insufficiently reviewed here. From this paragraph, it appears that the hybridoma strategy with a single clinical trial seems to be the only approach. Please consider also to mention strategies based on monocyte derived DCs, e.g., Khoury, Cancer 2017, Anguille Blood 2017, Lichtenegger, Clin Transl Immunology 2020.

Response: We have amended this section by providing additional reviewed data on dendritic cell-based vaccines, considering the suggested references. 

  • References: It would be helpful for the reader if conference abstracts would be marked as such compared to full publications.

Response: We have listed the conference abstracts in an additional file. Technically it was not possible for us to directly mark these references in the manuscript and would therefore kindly ask the editors to mark the corresponding conference abstracts in the references listed in the manuscript.

Reviewer 3 Report

The Manuscript by Sendker and colleagues is well written and contains adequate Figures, of which the resolution was somewhat limited in the current PDF. It addresses all aspects of immunotherapy in AML comprehensively with a specific focus on childhood AML and its challenges.

Major comments:

Despite the title of the review contains the words ‘immune microenvironemt’ the later sections on antibodies, cellular and cytokine therapies lose this focus and become more general in terms of how immunotherapy works. This connection is better in the first 30% of the paper. Otherwise, the title needs to be tuned towards more general aspects of immunotherapy in AML.

When the authors discuss immune-checkpoint blockade in 225 ff, they should provide some arguments why this strategy has not panned out so far in AML: Lower Mutational burden? Not sufficient antileukemic T cell responses which may be unleashed ? Systemic vs. local disease and the likelihood to raise an immune response?

Minor comments:

In the section about immune escape due to HLA loss etc. the paper on downregulation of MHC class II expression by Christopher et al, NEJM 2018: Immune Escape of Relapsed AML Cells after Allogeneic Transplantation | NEJM should be added.

There seems to be a Typo in line 119: Galectin 9 instead of Galactin.

Minor language editing is recommended.

Author Response

Reviewer 3

Review Report (Round 1)

The Manuscript by Sendker and colleagues is well written and contains adequate Figures, of which the resolution was somewhat limited in the current PDF. It addresses all aspects of immunotherapy in AML comprehensively with a specific focus on childhood AML and its challenges.

Response: Thank you. The figures will be provided in a higher resolution as additional files.

Major comments:

Despite the title of the review contains the words ‘immune microenvironemt’ the later sections on antibodies, cellular and cytokine therapies lose this focus and become more general in terms of how immunotherapy works. This connection is better in the first 30% of the paper. Otherwise, the title needs to be tuned towards more general aspects of immunotherapy in AML.

Response: We greatly acknowledge this valuable comment. We consider i.e., T-cells, NK-cells and dendritic cells as well as molecular factors such as different cytokines as pivotal component of the immune microenvironment. These microenvironmental factors essentially contribute to leukemogenesis and can be harnessed as immunotherapeutic approaches reviewed in the manuscript, i.e., using dendritic cell-based vaccines or oncolytic viruses, which causes microenvironmental changes thus ameliorating anti-leukemic immunity in the microenvironment to fight leukemic cells. In this regard the reviewed therapeutic approaches are based on pathological processes focusing malignantly altered immunological cellular and non-cellular components in the immune microenvironment in AML, which can be redirected using the mentioned therapeutic strategies. In case the reviewer still does not consider the title appropriate and disagree with the title, we can change it into “Restoring Immune surveillance in Acute Myeloid Leukemia” or “Redirecting immunity in a malignant microenvironment in Acute Myeloid Leukemia”, emphasizing the broad spectrum of the herein reviewed immunotherapeutic approaches in AML.

When the authors discuss immune-checkpoint blockade in 225 ff, they should provide some arguments why this strategy has not panned out so far in AML: Lower Mutational burden? Not sufficient antileukemic T cell responses which may be unleashed? Systemic vs. local disease and the likelihood to raise an immune response?

Response: We have modified this section and added additional statements, addressing this interesting question. Specifically, we discussed mutational burden and limited T-cell recognition, the impact of the immunosuppressive BMM on therapeutic efficaciousness of CPI as well as the stated relatively low baseline expression of checkpoint molecules and distinct co-expression patterns. Further discussion would go beyond the scope of this review.

Minor comments:

In the section about immune escape due to HLA loss etc. the paper on downregulation of MHC class II expression by Christopher et al, NEJM 2018: Immune Escape of Relapsed AML Cells after Allogeneic Transplantation | NEJM should be added.

Response: Thank you for the kind advice, we have added the publication.

There seems to be a Typo in line 119: Galectin 9 instead of Galactin.

Response: We have corrected the typo, as requested.

Minor language editing is recommended.

Response: The review has been edited by a native speaker as stated in acknowledgement.

Reviewer 4 Report

This review manuscript is well-organized and well-written.  Almost all related literatures were included.  The literatures on immune bone marrow microenvironment are very well summarized and described which provide strong background information to support the targeted immunotherapies that are discussed in the 2nd section of the manuscript. The review topic is very important which provide good information for both basic and clinical researchers to update the advance of the related field.

However, there is a significant prognostic difference between pediatric and adult AML patients.  It is unknown whither immune environment contributes to such clinical outcome difference.  It will be better if the authors can provide a short discussion on such issue by comparing the immune environment between pediatric and adult patients. 

In addition, it will be helpful if the authors can also provide a table to summarize the results, conclusions and side effects of the related clinical trials.

Author Response

Reviewer 4

Review Report (Round 1)

This review manuscript is well-organized and well-written.  Almost all related literatures were included.  The literatures on immune bone marrow microenvironment are very well summarized and described which provide strong background information to support the targeted immunotherapies that are discussed in the 2nd section of the manuscript. The review topic is very important which provide good information for both basic and clinical researchers to update the advance of the related field.

However, there is a significant prognostic difference between pediatric and adult AML patients.  It is unknown whither immune environment contributes to such clinical outcome difference.  It will be better if the authors can provide a short discussion on such issue by comparing the immune environment between pediatric and adult patients. 

Response: Thank you for the precious comments. Especially the last point is very exciting for us, regarding the known prognostic differences in adult and pediatric AML as well as facing the great heterogeneity of AML, which not only imply genetic landscape but may also imply microenvironmental changes. However there seems to be a significant lack of knowledge since we did not find suitable publication on developmental changes of the immuno-microenvironment in adults versus children and its prognostic impact. However, examining age-related differences of the microenvironmental immunity in pediatric and adults AML could be a great topic for a huge research project.

In addition, it will be helpful if the authors can also provide a table to summarize the results, conclusions, and side effects of the related clinical trials.

Response: Thank you for this suggestion. We have created a tabular overview, which summarizes the main important data including the study-settings, main results and relevant treatment related adverse events of selected trials.

Round 2

Reviewer 2 Report

Thank you for considering my suggestions.

Reviewer 3 Report

Points raised by the reviewers have been answered adequately. In fact, the alternative headings sugested by the reveiwers seem to be better fits.